# Grounding Robot Policies with Visuomotor Language Guidance

## Abstract

Recent advances in the fields of natural language processing and computer vision have shown great potential in understanding the underlying dynamics of the world from large-scale internet data. However, translating this knowledge into robotic systems remains an open challenge, given the scarcity of human-robot interactions and the lack of large-scale datasets of real-world robotic data. Previous robot learning approaches such as behavior cloning and reinforcement learning have shown great capabilities in learning robotic skills from human demonstrations or from scratch in specific environments. However, these approaches often require task-specific demonstrations or designing complex simulation environments, which limits the development of generalizable and robust policies for new settings. Aiming to address these limitations, we propose an agent-based framework for grounding robot policies to the current context, considering the constraints of a current robot and its environment using visuomotor-grounded language guidance. The proposed framework is composed of a set of conversational agents designed for specific roles—namely, high-level advisor, visual grounding, monitoring, and robotic agents. Given a base policy, the agents collectively generate guidance at run time to shift the action distribution of the base policy towards more desirable future states. We demonstrate that our approach can effectively guide manipulation policies to achieve significantly higher success rates both in simulation and in real-world experiments without the need for additional human demonstrations or extensive exploration. Project videos at https://sites.google.com/view/motorcortex/home.

## 1 Introduction

In recent years, the advent of foundation models, such as large-scale pre-trained language models (LLMs) and visual language models (VLMs), has shown great capabilities in understanding context, scenes, and the underlying dynamics of the world. Furthermore, emergent capabilities such as in-context learning have shown great potential in the transfer of knowledge between domains, e.g., via few-shot demonstrations or zero-shot inference. However, the application of these models to robotics is still limited, given the intrinsic complexity and scarcity of human-robot interactions and the lack of large-scale datasets of human-annotated data or demonstrations.

Most approaches that target using LLMs and VLMs in robotics often fall in one of two directions. Fine-tuning the models (Brohan et al., 2023; Ahn et al., 2022), which have shown great real-world capabilities, at the expense of large platform-dependent datasets. Even though several efforts try to overcome the data availability problem (Collaboration et al., 2023), the scarcity of robots and the broad range of robotics skills, make this process extremely taxing. A second line of work relies on using code as an interface between the language models and the robotic systems (Liang et al., 2023; Vemprala et al., 2023), which although leverages the skilled coding capabilities of this type of model, is highly dependent on handcrafted functions to interface the platform actions and perception.

Unsupervised approaches in robotics have shown great progress in exploring large state spaces and learning common sets of behaviors in complex environments. However, most of these approaches aim to learn the underlying dynamics of the environments from scratch, taking a large amount of iterations to learn seemingly basic skills.

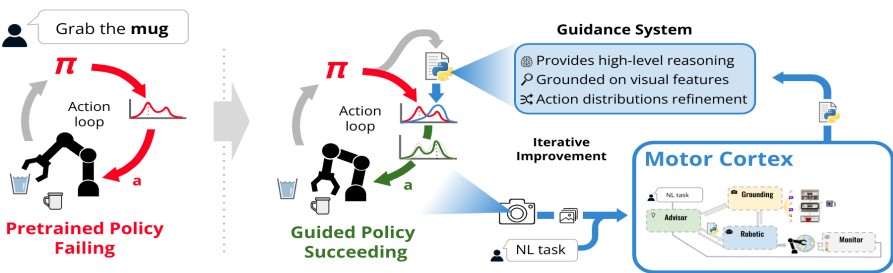

Figure 1: An overview of the proposed self-improving framework where the policy is updated by visuomotor-grounded guidance at test time.

In this context, we propose a cognitive approach motivated by how humans learn to improve performance. Exploration, abstraction, reasoning, and self-improvement are key components of human learning (Francis et al., 2022; Hu et al., 2023). In this paper, we aim to achieve self improvement by generating visuomotor guidances. We design an agent-based framework where a team of conversational agents with specific roles works together to refine the robot's base policy via grounded visuomotor guidance as shown in Figure 1. For instance, as illustrated in Figure 2, upon a request from the advisor agent, the grounding agent can look for objects by not only detection and tracking but also higher level reasoning to broaden the search, e.g., search inside a cupboard to find a mug; the robotic agent can reason about its own capabilities and constraints to provide feedback on the guidance generated by the advisor agent. Motivated by recurrent models, the progress of task performance is updated in a hidden state and passed down during the iterative guidance generation. The guidance is represented in the action distribution such that it can be easily combined with the base policy.

In our experiments on a set of benchmark tasks, we quantitatively show that our approach significantly improves the task success rate of two state-of-the-art robot policies. Our preliminary experiments on zero-shot learning show promising results where our approach enables the robot to learn to perform certain tasks from scratch without any demonstrations or other training material. We qualitatively demonstrate our approach on a real robot learning to perform composite manipulation tasks. Our contributions are as follows:

- We propose `g-MotorCortex`, an agentic robot policy grounding framework that can self-improve by generating the action-scoring guidance functions to update the action distribution of a base policy in an online manner. Our experiments in both simulated and real-world robot settings show significant performance improvements on both adapting existing policies and learning new skills from scratch.
- In order to enable robustness against cluttered and unseen environments, we propose a grounding agent that performs multi-granular object search, which enables flexible visuomotor grounding.
- Our approach shows a promising potential for learning skills from scratch after deployment and being generalizable across various base policy classes, tasks, and environments. To promote further research in this direction, we open source the code, models, and system prompts.

## 2 RELATED WORK

**Vision-Language Models for Robot Learning:** Several works explore the notion of leveraging pre-trained or fine-tuned Large Language Models (LLMs) and/or Vision-Language Models (VLMs) for high-level reasoning and planning in robotics tasks (Hu et al., 2023; Ahn et al., 2022; Liang et al., 2023; Huang et al., 2022; Singh et al., 2023; Huang et al., 2023a; Ha et al., 2023; Ding et al., 2023; Michał et al., 2024; Ma et al., 2023; Li et al., 2024; Mu et al., 2024)—typically decomposing high-level task specification into a series of smaller steps or action primitives, using system prompts or in-context examples to enable powerful chain-of-thought reasoning techniques. This strategy of encouraging models to reason in a stepwise manner before outputting a final answer has led to significant performance improvements across several tasks and benchmarks (Hu et al., 2023). Despite these promising achievements, these approaches rely on handcrafted primitives (Ahn et al., 2022; Huang et al., 2022; Liang et al., 2023; Michał et al., 2024), struggle with low-level control, or

require large datasets for retraining. furthermore, various approaches that leverage VLMs for robot learning suffer from a granularity problem when using off-the-shelf models in a single-step/zero-shot manner (Huang et al., 2023b) or are unable to perform failure correction without costly human intervention (Huang et al., 2022; Michał et al., 2024; Liang et al., 2024; Huang et al., 2023b). In contrast, our framework bridges high-level reasoning with low-level control, by leveraging an agentic framework for online modification of a base policy's action distribution at test-time, without requiring human feedback or datasets for fine-tuning. Moreover, we mitigate the granularity problem by proposing a flexible and recurrent way of using VLMs to query open-vocabulary perception models.

**Agent-based VLM frameworks in Robotics:** Rather than using single VLMs in an end-to-end fashion, which might incur issues in generalization and robustness, various works have sought to orchestrate multiple VLM-based agents to work together in an interconnection multi-agent framework. Here, multiple agents can converse and collaborate to perform tasks, yielding improvements for the overall framework in online adaptability, cross-task generalization, and self-supervision (Xu et al., 2023; Zhang et al., 2024; Parakh et al., 2023). These *agentic* frameworks have provided possibilities for enabling the identification of issues in task execution, providing feedback about possible improvements. Challenges remain, however, in that this feedback is often not sufficiently grounded on the spatial, visual, and dynamical properties of embodied interaction to be useful for policy adaptation; instead, the generated feedback is often too high-level or provides merely binary signals of success or failure.

**Self-guided Robot Failure Recovery:** Guan et al. (2024) offer an analysis of frameworks for leveraging VLMs as behavior critics. Some approaches have explored integrating such pre-trained models to improve the performance of reinforcement learning (RL) algorithms. For instance, Ma et al. (2023) use LLMs in a zero-shot fashion to design and improve reward functions, however this approach relies on human feedback to generate progressively human-aligned reward functions and further requires simulated retraining via RL, with high sample-complexity. On the other hand, Rocamonde et al. (2023) avoid the need for explicit human feedback by directly using a VLM (CLIP) to compute the rewards to measure the proximity of a state (image) to a goal (text description), enabling gains in sample-efficiency for guiding a humanoid robot to perform various maneuvers in the MuJoCo simulator. A limitation of this approach, however, lies in the difficulty of generating rewards for long-horizon or multi-step tasks, which are characteristic of tasks involving complex agent-object interactions. Liu et al. (2023b) present a framework for detecting and analyzing failed executions automatically. However, their system focuses on explaining failure causes and proposing suggestions for remediation, as opposed to also performing policy correction. In this paper, we propose a framework that directly adapts a base policy's action distribution, during deployment, without requiring additional human feedback.

## 3 GROUNDING ROBOT POLICIES WITH GUIDANCE

### 3.1 PROBLEM FORMULATION

We consider a pre-trained stochastic policy $\pi : O \times S \to A$ that maps observations $o_t$ and robotic states $s_t$ to action distributions $a_{\pi,t}$, at each time step $t$. Our objective is to generate a guidance distribution $g_t$ that, when combined with this base policy, enhances overall performance during inference without requiring additional human demonstrations or extensive exploration procedures. Specifically, we aim to develop a modified policy $\pi_{\text{guided}} : O \times S \to A$ that achieves better performance on tasks where the original policy $\pi$ struggles. We define this new policy as follows:

$$\pi_{guided}(a_{g,t}|o_t, s_t) = \pi(a_{\pi,t}|o_t, s_t) * G(a_{\pi,t}|o_t, s'_{t+1}), \tag{1}$$

where $G : A \times O \times S \to [0, 1]$ is a guidance function that maps observation $o_t$, action $a_t$, possible future state $s'_{t+1}$ into a guidance score $g_t$. The '$*$' operator here denotes the operation of combining both distributions conceptually, which we explore in detail in Section 3.4. For the scope of this project, we assume that a dynamics model $\mathcal{D} : S \times A \to S$ is available, which can forecast possible future states of the robot $s'_{t+1} = \mathcal{D}(s_t, a_{\pi,t})$ given the current state $s_t$ and action $a_{\pi,t}$.

Focusing on leveraging the world knowledge of Vision Language Models, while avoiding adding latency to the action loop, we choose to express these guidance functions as Python code. By integrating these code snippets into action loop of the base policies, we eliminate the need of time-consuming queries to large reasoning models. Samples of the format and content of the guidance functions generated by the framework are presented in the Appendix A.2.

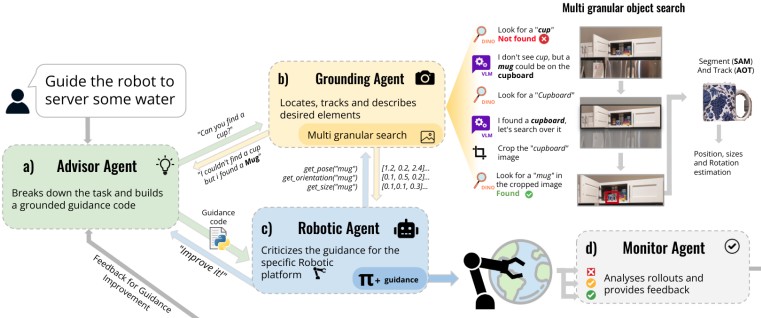

Figure 2: Information flow between the agents to produce a guidance code. **a)** The advisor agent orchestrates guidance code generation by collaborating with other agents and using their feedback to refine the generated code. **b)** The grounding agent uses segmentation and classification models to locate objects of interest provided by the advisor, reporting findings back to the advisor. **c)** The robotic agent uses a Python interpreter to test the code for the specific robotic platform and judge the adequacy of the code. **d)** The monitor agent analyses the sequence of frames corresponding to the rollout of the guidance and give feedback on potential improvements.

### 3.2 A MULTI-AGENT GUIDANCE FRAMEWORK FOR SELF IMPROVEMENT

In order to generate the guidance function $G$, we leverage a group of conversational agents empowered with visual grounding capabilities and tool usage. Illustrated in Figure 2, the framework is composed of four main agents: an Advisor Agent, the Grounding Agent, the Monitor Agent, and the Robotic Agent. We provide system prompt samples in Appendix A.1.

*Advisor Agent.* A Vision Language Model is responsible for breaking down the task and communicating with the other agents to generate a sound guidance function for a given task.

*Grounding Agent.* A Vision Language Model that iteratively queries the free-form text segmentation models to locate (Cheng et al., 2023a; Liu et al., 2023a), track (Cheng et al., 2023b), and describe elements relevant to the task execution.

*Monitor Agent.* Responsible for identifying the causes of the failures in the unsuccessful rollouts, the Monitor Agent consists of a Vision Language model equipped with a key frame extractor.

*Robotic Agent.* Language Model equipped with descriptions of the robot platform, a robot's dynamics model and wrapper functions for integration with the base policy. It criticizes the provided guidance functions to reinforce its relevance to the task and alignment with the robot's capabilities.

### 3.3 GUIDANCE PROCEDURE

The conversational agents interact with each other through natural language and query their underlying tools to iteratively produce a guidance code tailored to the task in hand, the environment, and the robot's capabilities. The information flow between these agents is depicted by Figure 2.

For a given task expressed in natural language and an image of the initial state of the environment, the *Advisor Agent* uses Chain-of-Tought (Wei et al., 2023) strategy to generate a high-level plan of the steps necessary to accomplish the task. Being able to query a *Grounding Agent* and the *Robotic agent*, the Advisor is able to collect relevant information about trackable objects and elements in the environment, as well as the capabilities and limitations of the robotic platform.

For a given plan and list of relevant objects required for the task completion, the Grounding Agent uses grounding Dino (Liu et al., 2023a) and the Segment Anything Model (SAM) (Cheng et al., 2023a) to locate the elements across multiple granularities and levels of abstraction. For instance, if an object is not immediately found, the agent will actively look for semantically similar objects or will look for higher-level elements that could encompass the missing object. For example, if the object "cup" is to be located, and it could not be immediately found, the agent could search for similar object like a "mug". If it still struggles to locate it, the agent could search for a "shelf" and then try to find the "cup" or "mug" in the cropped image of the "shelf". If an object is found, it is added to a tracking system (Yang & Yang, 2022). This process enhances the Segment and Track Anything (Cheng et al., 2023a) approach with flexible multi-granular search. The object statuses are

reported back to the Advisor Agent, which iterates on the action plan or proceeds with the generation of a guidance function grounded on the trackable objects located.

The Robotic agent acts as a critic to improve the guidance function generated. Equipped with a Python interpreter and details of the base policy's action space, the agent can evaluate the guidance function in terms of feasibility and relevance to the robot's capabilities. Once a function suffices the system's requirements, it is saved to be used in the action loop, in combination with the dynamics model, to provide a guidance score for possible actions sampled from the base policy.

After the execution of a rollout and the identification of failure in the task completion, the Monitor Agent is triggered to analyze the causes of the failure. By extracting key frames from the rollout video using PCA (Maćkiewicz & Ratajczak, 1993) and K-means clustering, the agent can feed a relevant and diverse set of images to the Visual Language Model prompted to access the failure causes. In the iterative applications of our framework, the Monitor Agent provides this feedback to the Advisor Agent, which can use this information to refine the guidance functions generated in the previous iterations.

**Temporal-aware Guidance Functions.** Inspired by recurrent architectures, we instruct the agents to generate guidance function conditioned on a customizable hidden state ($h_t$) expressed as an optional dictionary parameter as shown in the following example:

```python
# Guidance function example in the context of grabbing a mug
def guidance_code(state,
    hidden_state={"mug_reached": False,"mug_grabbed":False}):
    #available grounding functions
    #x,y,z = get_pose("mug")
    #h,w,d = get_size("mug")
    #rx,ry,rz = get_orientation("mug")
    ...
    return score, new_hidden_state
```

The idea of using abstraction in a hidden state has proven to significantly improve the guidance performance, allowing the guidance functions to keep track of the task progress and adapt the guidance to longer horizon tasks. The guided policy can thus be written as:

$$\pi_{guided}(a_{g,t}|o_t, s_t, h_t) = \pi(a_{\pi,t}|o_t, s_t) * G(a_{\pi,t}|o_t, s'_{t+1}, h_t). \tag{2}$$

The complete guidance procedure is summarized in Algorithm 1. Note that we refer to the self-orchestrated conversation between the agents which yields the guidance code as the function `Generate_Guidance`. For a closer look at the chain of thought employed by each agent please refer to Appendix A.1.

### 3.4 GUIDANCE AND POLICY INTEGRATION

Aiming to guide a wide range of policies, our framework is designed to work both with continuous and discrete action spaces. In this section, we discuss the operation of combining the guidance function with the base policy's action distributions. Furthermore, we discuss how deterministic regression models can be adapted to work with our framework.

**Action-space Adaptation.** We assume the availability of a dynamics model $\mathcal{D}$ that can forecast possible future states of the robot given a possible action $a'_{\pi,t}$. In the manipulation domain, a dynamics model is often available in the form of a forward kinematics model, a learned dynamics model, or a simulator. Oftentimes, the action space $A$ of policies them-self is the same as the robot's state $S$ either being or joint angles of the robot or the gripper's end-effector pose. For the last cases, where both the action and state space are expressed in $SE(3)$ integrating the guidance function with a base policy would only require a multiplication of the guidance scores with the action probabilities of the base policy. In other scenarios, adapting the robot's action and state space to match the representation of the visual cues (position, orientation, and size) would be required.

Considering the visual grounding, the action space and the state space share the same representation ($SE(3)$), the operation to combine the guidance function with the base policy can be expressed as an element-wise weighted average:

$$\pi_{guided} = (1 - \alpha)\pi \cdot \alpha G, \tag{3}$$

where $\alpha \in [0, 1]$ represents the percentage of guidance applied with respect to the base-policies distribution and is here denoted as *guidance factor*.

---

**Algorithm 1** Guidance procedure

  **Input**
  $\pi$: Base policy
  $\mathcal{D}$: Dynamics Model
  env: Environment

1: **for** each episode **do**
2:     Obtain observation and initial state $o_t, s_t \leftarrow$ env.init
3:     Generate guidance function with our framework $G \leftarrow$ Generate_Guidance$(o_t, s_t)$
4:     Initialize hidden states $h_t \leftarrow$ Generate_Hidden_State$(G(o_t, s_t))$
5:     **for** each time step $t$ **do**
6:         Sample $n$ actions from the policy $\mathcal{A}_{\pi,t} \leftarrow \{\pi(o_t, s_t)^i\}_{i=0}^n$
7:         Get action probabilities $\pi_t \leftarrow \pi(\mathcal{A}_{\pi,t}|o_t, s_t)$
8:
9:         Infer possible future states $\mathcal{S}_{\pi,t} \leftarrow \mathcal{D}(s_t, \mathcal{A}_{\pi,t})$
10:       Compute the guidance for the sampled possible future states $G_{\pi,t} \leftarrow G(o_t, \mathcal{S}_{\pi,t}, h_t)$
11:       Normalize $G_{\pi,t}$
12:       Combine distributions $\pi_{\text{guided},t} \leftarrow \pi_t * G_{\pi,t}$
13:       Select the best action $a_t \leftarrow \mathcal{A}_{\pi,t}[\text{argmax}(\pi_{\text{guided},t})]$
14:       $o_t, s_t \leftarrow$ env.step$(a_t)$        ▷ Execute $a_t$, update state $s_{t+1}$ and observation $o_{t+1}$
15:       Update hidden states $h_t \leftarrow$ Generate_Hidden_State$(G(o_t, s_t, h_t))$

---

**Adaptation of Regression Policies.** To properly leverage the high-level guidance expressed in the guidance functions and the low-level capabilities of the base policy, it is desired that the policy's action space be expressed as a distribution. In the case of regression policies that do not provide uncertainty estimates, several strategies can be employed to infer the action distribution. One common approach is to assume a Gaussian distribution centered at the predicted value and compute the variance using ensembles of models trained with different initialization, different data samples, or different dropout seeds or different checkpoint stages (Abdar et al., 2021). Other strategies to infer the distributions of the model include using bootstrapping, Bayesian neural networks, or using a mixture of Gaussians (Mena et al., 2021).

### 3.5 Learning New Robot Skills from Scratch

We note that this framework can enable robots to acquire new skills from scratch through zero-shot learning or self-improvement via iterative guidance updates. By leveraging the system's ability to ground guidance in the visual features of the environment, the system can perform tasks without prior training. Applying 100% guidance over untrained policies, the robot explores the environment with purpose, learning basic skills and refining them iteratively to improve success. In section 4, we provide an analysis of the system's performance in such challenging scenarios.

## 4 Experiments & Results

### 4.1 Experimental Setup

**Task Definitions:** We demonstrate the efficacy of `g-MotorCortex`, in simulation on the RL-Bench benchmark (James et al., 2020) and on two challenging real-world tasks. For the real-world setup, we use the UFACTORY Lite 6 robot arm as the robotic agent and, as the end-effector, we use the included UFACTORY Gripper Lite, a simple binary gripper. The arm is mounted on a workbench. For perception, we use a calibrated RGB-D Camera, specifically the Intel RealSense Depth Camera D435i. All experiments were conducted on a desktop machine with two (2) NVIDIA RTX 3090 GPU, 64GB of RAM, and an Intel i9-10900K CPU.

- *(Sim)*: **RL-Bench:** We consider 10 tasks on the RL-Bench benchmark (James et al., 2020) using a single RGBD camera input, as described in the *GNFactor* setup (Ze et al., 2024).
- *(Sim)*: **RL-Bench, learning from scratch:** Aiming to explore the capabilities of `g-MotorCortex` on learning new skills from scratch, we selected 4 challenging tasks from the RL-Bench benchmark: *turn tap*, *push buttons*, *slide block to color target*, *reach and drag*.

- *(Real)*: **Sequenced Multi-button Press:** Here, the agent must use its end-effector to press multiple real buttons on a workspace, in a particular order. We designed this task to evaluate whether the proposed framework is capable of improving pre-trained robot policies in performing challenging tasks, without access to human demonstrations.
- *(Real)*: **Reach for chess piece:** Given a cluttered scene with many similar objects, we want to evaluate if the multi-granular perception framework can effectively guide the agent to identify and reach for the appropriate target. We implement this perceptual grounding and reaching task on a standard chessboard, where the agent must identify and reach for one of the chess pieces specified by natural language instruction.

**Base Policies:** We evaluate the effectiveness of our guidance framework using different base policies, *Act3D* (Gervet et al., 2023), *3D Diffuser Actor* (Ke et al., 2024), and a RandomPolicy. All policies plan in a continuous space of translations, rotations, and gripper state ($SE(3) \times \mathbb{R}$), however they utilize different inference strategies:

- *Act3D* samples waypoints in the Cartesian space ($\mathbb{R}^3$) and predicts the orientation and gripper state for the best scoring sampled waypoint, combining a classification and regression strategies into a single policy.
- *3D Diffuser Actor*, on the other hand, uses a diffusion model to compute the target waypoints and infers the orientation and gripper state from the single forecast waypoint, thus tackling the problem as a single regression task.
- *RandomPolicy* denotes any of the former frameworks that has not been trained for a specific task, therefore the weights are randomly initialized.

The fundamentally different types of policies' outputs make them a great use case for our policy guidance framework. Furthermore, the common representation of the action and state spaces of both policies ($SE(3) \times \mathbb{R}$) provides a straight forward integration with our grounding models.

As described in Section 3.4, the regression components of the policies require an adaptation to transform the single predictions of the model into a distribution over the action space. For the sake of simplicity, we assume a Gaussian distribution over the action space, with the mean centered on the predicted values and the standard deviation fixed on a constant value. The outputs of the classification component of Act3D (waypoint positions) were directly considered as samples of a distribution over the Cartesian space ($\mathbb{R}^3$).

The integration of base policies with the guidance distributions was performed by applying a weighted average parameterized by $\alpha$ as shown in Equation 3.

### 4.2 EXPERIMENTAL EVALUATION

Our experimental evaluation aims to address the following questions: (1) Does `g-MotorCortex` improve the performance of pre-trained base policies on specific robotics tasks and environments without additional human demonstrations? (2) Does the proposed multi-granular perception capabilities effectively guide the policy in challenging cluttered environments? (3) Does `g-MotorCortex` enable policies to learn new skills from scratch? (4) What is the effect of guidance on expert versus untrained policies?

**Does `g-MotorCortex` improve the performance of pre-trained base policies on specific robotics tasks and environments without additional human demonstrations?** We first assess the effect of the proposed guidance on the *Act3D* and *3D Diffuser Actor* baselines following the *GN-Factor* (Ze et al., 2024) setup, which consists of a single RGBD camera and table-top manipulator performing 10 challenging tasks with 25 variations each. Guidance is iteratively generated for the failure cases. For the failed rollouts, our policy improvement framework ran for 5 iterations. As displayed by Table 1, the framework was able to improve the success rate of the base policy on most of the tasks, with the best results achieved by using 1% guidance. The low amount of guidance has shown to be enough to bend the action distribution to the desired direction, while still preserving the low-level nuances captured by the base policy. This suggests that `g-MotorCortex` is capable of improving base policies by adding abstract understanding and grounding of the desired task, while preserving the low-level movement profiles captured by the original policies.

**Does the proposed multi-granular perception capability effectively guide the policy in challenging, cluttered environments?** In real-world experiments, we qualitatively demonstrate the fine-grained detection capabilities of `g-MotorCortex` by tasking it with reaching for a white

Table 1: Performance improvement on the RL-Bench (James et al., 2020) benchmark, by applying 5 iterations of guidance improvement over unsuccessful rollouts.

| Model | turn tap | open drawer | sweep to dustpan of size | meat off grill | slide block to color target | push buttons | reach and drag | close jar | put item in drawer | stack blocks | Avg. Success |
|---|---|---|---|---|---|---|---|---|---|---|---|
| Act3D 25 demos/-task | 76 | 76 | 96 | 64 | 92 | 84 | 96 | 48 | 60 | 0 | 69.2 |
| +1% guidance | 80 (+4) | 96 (+20) | 96 | 84 (+20) | 92 | 84 | 100 (+4) | 84 (+36) | 80 (+20) | 8 (+8) | **80.4 (+11.2)** |
| +10% guidance | 88 (+12) | 100 (+24) | 96 | 88 (+24) | 92 | 84 | 100 (+4) | 60 (+12) | 80 (+20) | 0 | 78.8 (+9.6) |
| Act3D 10 demos/-task | 32 | 60 | 84 | 16 | 60 | 72 | 68 | 32 | 44 | 8 | 47.6 |
| +1% guidance | 44 (+12) | 88 (+28) | 88 (+4) | 24 (+8) | 68 (+8) | 72 | 76 (+8) | 52 (+20) | 60 (+16) | 8 | **58 (+10.4)** |
| +10% guidance | 44 (+12) | 64 (+4) | 84 | 20 (+4) | 68 (+8) | 76 (+4) | 76 (+8) | 40 (+8) | 56 (+12) | 8 | 53.6 (+6) |
| Act3D 5 demos/task | 24 | 0 | 84 | 4 | 8 | 32 | 8 | 8 | 12 | 0 | 18 |
| +1% guidance | 48 (+24) | 16 (+16) | 84 | 8 (+4) | 12 (+4) | 40 (+8) | 24 (+16) | 20 (+12) | 20 (+8) | 0 | **27.2 (+9.2)** |
| +10% guidance | 24 | 0 | 84 | 8 (+4) | 12 (+4) | 44 (+12) | 20 (+12) | 8 | 20 (+8) | 0 | 22 (+4) |
| Diffuser actor 5 demos/ task | 24 | 64 | 40 | 28 | 44 | 68 | 40 | 24 | 44 | 0 | 37.6 |
| +1% guidance | 40 (+16) | 92 (+28) | 64 (+24) | 40 (+12) | 44 | 68 | 52 (+12) | 24 | 84 (+40) | 0 | **50.8 (+13.2)** |
| +10% guidance | 40 (+16) | 84 (+20) | 52 (+12) | 28 | 52 (+8) | 68 | 48 (+8) | 32 (+8) | 76 (+32) | 0 | 48 (+10.4) |

knight chess piece in a cluttered chess board. Figure 3 shows the roll-out of the first guidance iteration over an untrained policy, displaying the initial and final time steps of the task. The accompanying heatmaps illustrate the distributions of the original untrained policy (Diffuser heatmaps) and the guided policy (Guidance heatmaps). Additionally, the multi-granular search results highlight the steps taken by the grounding agent to locate the target piece. After initially failing to detect the white knight directly, the agent successfully identifies the chessboard and then focuses within that region to locate the target piece. These findings demonstrate that `g-MotorCortex` effectively leverages a semantic understanding of scene components to guide the policy towards successful task execution.

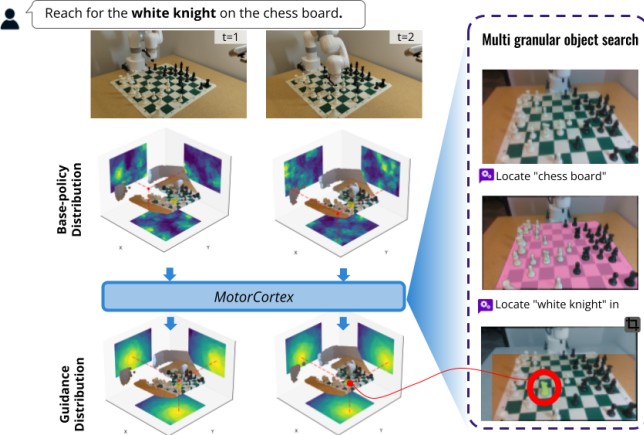

Figure 3: Real-world results for learning skills from scratch on the Lite6 chess task. The top row shows an external view of the robot performing the tasks. The second row depicts the action heat map given by the random diffuser policy at the first and last time step. The bottom row depicts the corresponding heat maps generated after applying the guidance. We show it can successfully guide the action towards the desired object. On the right, we show a breakdown of the multi-granular search performed by other **grounding agent** to locate the white knight; we disambiguate the scene by searching in parent objects and constraining the search to semantically relevant areas.

**Does `g-MotorCortex` enable policies to learn new skills from scratch?** We evaluated the performance of the framework on learning new skills from scratch on 4 tasks of the RL-Bench benchmark: *turn tap*, *push buttons*, *slide block to color target*, *reach and drag*. In this setup, we initialized the Act3D policy with random weights and applied 100% of guidance ($\alpha = 1.0$) over the policy for the x,y, and z components. Only leveraging the waypoint sampling mechanism of Act3D and overwriting its distribution with the values queried from the guidance functions generated. The results show that the framework is capable of learning new skills from scratch, achieving a higher success rate than the base policy pre-trained on 5 demonstrations/tasks for the tasks "turn tap" and "push buttons" tasks. When utilizing only the untrained Act3D policy (without guidance) the policy achieved 0 success rate on the tasks. Figure 4 demonstrates the iterative improvement of our guid-

ance framework. Updating the guidance code generated for each failed rollouts from the previous iteration. Policy rollouts are provided in Figure 7.

It is worth mentioning that a few variations of the simulated tasks "turn tap" and "reach and drag", which seemly would require a precise orientation control of the manipulator, could be solved by guiding only the Cartesian components of the police's output. For these variations, a qualitative analysis shows that successful roll-outs could be achieved by tapping the end-effector on the target objects.

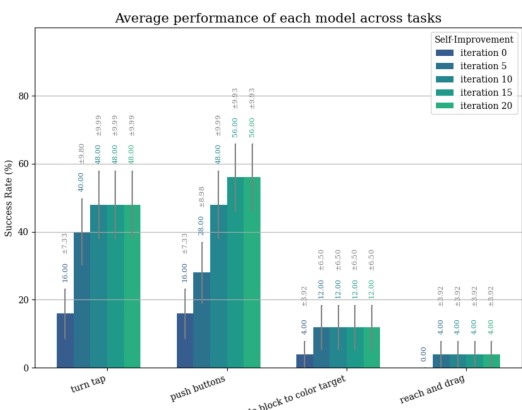

Figure 4: Performance of our framework on learning new skills from scratch (guidance over untrained policies), and iteratively improving the guidance functions generated.

We perform a similar experiment in the real-world settings. Here, we run the framework only relying on the action distribution given by the guidance code ($\alpha = 1.0$), using a Random-Policy as the base policy. We first consider the task of pressing colored buttons in a given sequence; using toy buttons made out of acrylic and paper as a prop. Figure 5 shows the roll-out corresponding to the first iteration for this task, along with heat maps depicting a projection of the output action distribution around the point of maximum. In this zero-shot scenario, `g-MotorCortex` has proven to correctly guide the robot to the desired objects preserving the prompted order; however, it struggles to capture low-level nuances of the movement, such as the appropriate pressing force and proper approach of the buttons. A simulated version of this experiment is shown in the appendix (Figure 7), demonstrating how combining the guidance with a pre-trained policy can mitigate this behavior.

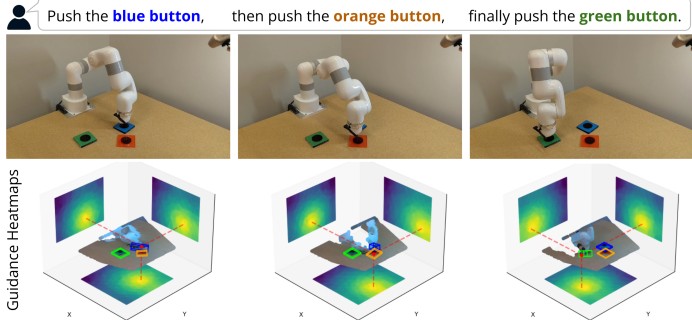

Figure 5: Real-world results for learning skills from scratch on the Lite6 on the multiple button press task. Each column represents a keyframe in the task rollout. The first row shows a third-person view of the robot's movement and the tabletop setting. The second shows the corresponding action distribution over the space generated by the guidance code, the red dot indicates the target waypoint for the end effector. We demonstrate that `g-MotorCortex` can guide a random base policy to successfully perform the desired task.

**What is the effect of guidance on expert versus untrained policies?** As discussed previously, `g-MotorCortex` can learn tasks from scratch using a random base policy with 100% of guidance ($\alpha$). Given that we are not affecting the policy, the product of the iterative learning from scratch is the generated guidance script which captures a high-level understanding of the task, e.g., spatial relationships, and task completion criteria, among others. We can qualitatively see the effect of the learning process by comparing the guidance scripts for different iterations of the same task. Appendix A.2 includes two samples of guidance code for the same task but on different iterations. We can see that the code corresponding to the second iteration is very simple and only accounts for Euclidean distance and button order; while by the fifth iteration considerations like orientation come into play.

On the other hand, applying the guidance to an existing expert policy aims to shift the action distribution to account for failure cases, like potentially out-of-distribution scenes. The goal here is to use the gained high-level understanding to aid the policy in task completion. Table 1 shows that adding too much weight to the guidance function can yield diminishing returns as it can overpower the nuanced low-level details from the expert policy: notice that the performance gain across the board is bigger using 1% of guidance.

## 5 CONCLUSION

**Summary:** In this work, we proposed `g-MotorCortex`, a novel framework for the self-improvement of embodied policies. Our self-guidance approach leverages the world knowledge of a group of conversational agents and grounding models to guide policies during deployment. We demonstrated the effectiveness of our approach in autonomously improving manipulation policies and learning new skills from scratch, in simulated RL-bench benchmark tasks and in two challenging real-world tasks. Our results show that the proposed framework is especially effective in improving the following high-level task structures and key steps to solve the task. This capability can be well suited for improving pre-trained policies that struggle with long-horizon tasks or for learning new simple skills from scratch.

**Limitations:** From an analysis of the guided rollouts, a few of the tasks variations proved challenging for the perception models used by the grounding agent, leading to false positives detections or failure to locate specific objects. This limitation was mainly observed in simulation task, were the graphics object representations, even though simplified, do not always match the representations used to train the object detection models. This limitation could be addressed by integrating more robust object detection models or verification procedures to ensure the correct detection of objects in the scene. Moreover, occasional inaccuracies on scene understanding by the Visual Language Model (VLM) have been observed, leading to the generation of inaccurate guidance codes and unexpected behaviors. Even though recent advances in large vision-language models have shown great potential in understanding the underlying dynamics of the world from large-scale internet data, translating this knowledge into out-of-distribution domain, such as robotics, while preventing hallucinations remains an open challenge.

**Future Works:** Regarding future works, we think that combining the proposed framework with fine-grained exploration techniques would allow the policy to explore in a targeted manner the low-level details of the task, while leveraging the high-level guidance provided by our framework. This may enrich the guidance codes with the necessary low-level details required to perform more complex tasks successfully.

Furthermore, the guidance function generation could be further improved by composing and adapting from a repository of successful guidance functions from previous experiences. This could be achieved by incorporating Retrieval Augmented Generation (RAG) (Lewis et al., 2021) into our multi agent framework. This modification could allow the guidance system to learn new simple skills from scratch by interacting with the environment and leveraging this collected knowledge to guide the policy more effectively.

Aiming to incorporate the knowledge captured by the guidance functions into the base policy, an experience replay and finetuning mechanisms could be incorporated into our current system. This modification could allow the framework to use past guided experiences to improve the base policy in a sample efficient manner. This could be achieved in a targeted matter by leveraging Low Rank Adaptation (LoRA) (Hu et al., 2021).

### 5.1 REPRODUCIBILITY STATEMENT

Intending to encourage other researchers to build upon the introduced framework, we take steps to ensure the usability and reproducibility of our work. The source code for `g-MotorCortex` is open-sourced and linked to on the project's website. We have provided dockerized scripts to facilitate the setup across different development environments. Additionally, in section A.1 we include the prompts used to configure each agent. The temperature of the model was set as zero to reduce variations in runs, as using fixed seeds for the experiments; more hyperparameter details are available in the open-sourced repository.

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

## A APPENDIX

### A.1 AGENT PROMPT CONFIGURATION

We provide the system prompts used to initialize each one of the agents. Note that for models relying on API calls, we use `gpt-4o-mini-2024-07-18`. The maximum number of tokens is set to 2000.

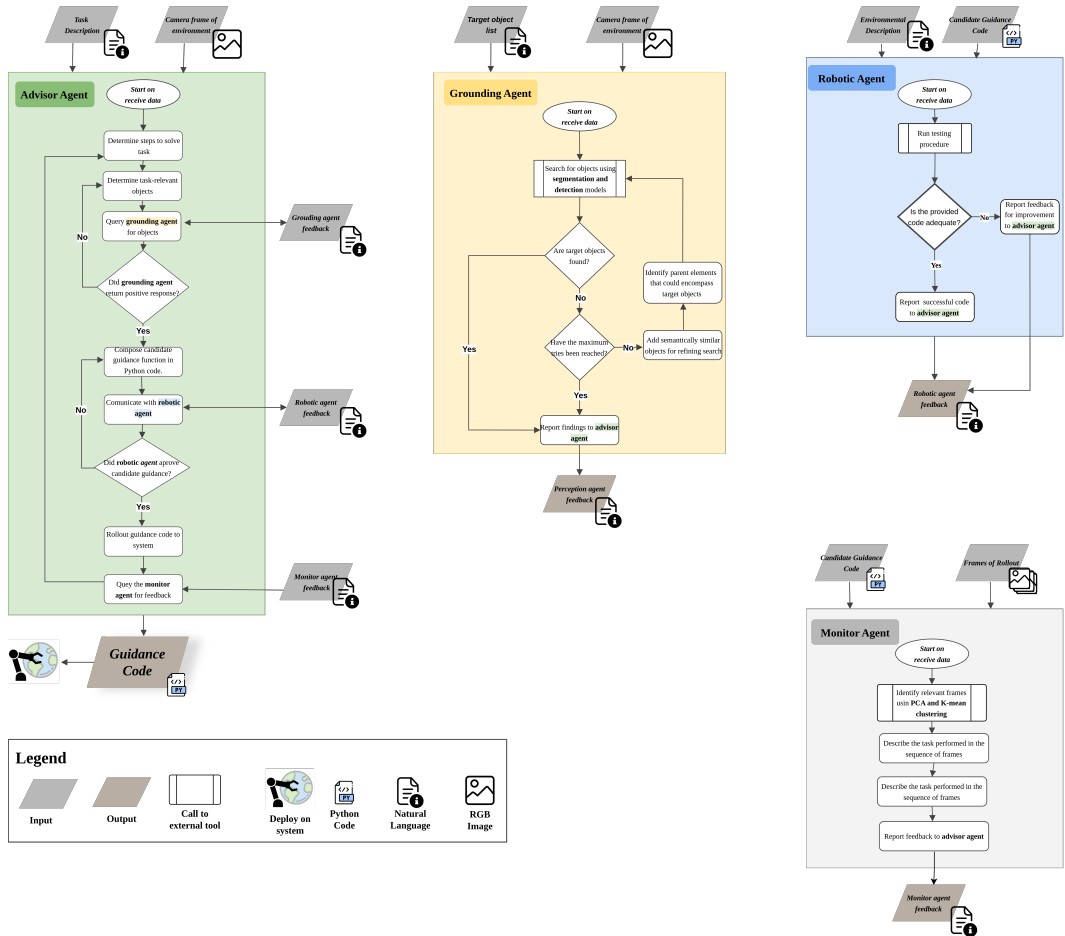

Figure 6: The agents in the g-MotorCortex framework are instances of large multimodal models that communicate with each other to produce a final guidance code; leveraging the reasoning capabilities of this type of model. This image exemplifies the chain of thought each agent is encouraged to follow, which in practice is encoded as a natural language prompt shown in Appendix A.1. The agents can call external tools to aid their analysis such as detection models and a Python interpreter for scrutinizing the code. The advisor agent acts as the main orchestrator, querying the other agents as necessary and generating and refining the guidance code with the provided feedback.

Robotic Agent Prompt

You are an AI agent responsible for controlling the learning process of a robot. You will receive Python code containing a guidance function that helps the robot with the execution of certain tasks. Your job is to analyze the environment and criticize the code provided by checking if the guidance code is correct and makes sense. You SHOULD NOT create any code, only analyze the code provided by the supervisor. Attend to the following:
- The score provided by the guidance function is continuous and makes sense.
- The task is being solved correctly.
- The code can be further improved.
- The states of the robot are being correctly expressed.
- The code correctly conveys the steps to solve the task in the correct order.
BE CRITICAL!
Make sure that the robot state is expressed as its end-effector position and orientation in the format by using the function test_guidance_code_format(). If the code

is not correct or can be further improved, provide feedback to the supervisor_agent and ask for a new code. Use 'NEXT: supervisor_agent' at the end of your message to indicate that you are talking with the supervisor_agent. If no code is provided, ask the supervisor_agent to generate the guidance code. If the code received makes sense and is correct, simply output the word 'TERMINATE'.

### Grounding Agent Prompt

You are a perception AI agent whose job is to identify and track objects in an image. You will be provided with an image of the environment and a list of objects that the robot is trying to find (e.g. door, handle, key, etc). With that, you can make use of the following function to try to locate the objects in the image: in_the_image(image_path, object_name, parent_name) - yes/no. If the object is not found it might be because the object was too small, too far, or partially occluded, in this case, try to find a broader category that could encompass the object. In this case, report the function call used followed by 'NEXT: perception agent' to look for the objects using similar object names or with a parent name that cloud encompasses the object. (e.g. first answer: 'in_the_image('door handle') - no NEXT: perception agent', second answer: 'in_the_image('door_handle', 'door') - no NEXT: perception agent', third answer: 'in_the_image('handle', 'gate') - yes. couldn't find a door handle but found a gate handle NEXT: supervisor_agent'). Report back to the supervisor agent in a clear and concise way if the objects were found or not. If an object was found using a parent name, report the parent name and the object name. Use 'NEXT: supervisor_agent' at the end of your message to indicate that you are talking with the supervisor_agent, or 'NEXT: perception_agent' to look further for the objects.

### Advisor Agent Prompt

You are a supervisor AI agent whose job is to guide a robot in the execution of a task. You will be provided with the name of a task that the robot is trying to learn (e.g. open door) and an image of the environment. With that, you must follow the following steps:
1- determine the key steps to solve the tasks.
2- come up with the names of features or objects in the environment required to solve the task.
3- check if the objects are present in the scene and can be detected by the robot by providing the image to the perception agent and asking the perception agent (e.g. 'Can you find the door handle?' wait for feedback), If the answer goes against of what you expected to repeat the steps 1 to 3.
4- Only proceed to this step after receiving positive feedback on 3. Write a Python code to guide the robot in the execution of the task. The output code needs to have a function that takes the robot's state as input (def guidance(state, previous_vars='condition1':False, ...):), queries the position of different elements in the environment (e.g get_position('door')) and outputs a continuous score for how close is the robot to completing the task (e.g. the robot is far away from the door the score should be low).

When writing the guidance function, you can make use of the following functions that are already implemented: get_position(object_name) -¿ [x,y,z], get_size(object_name) -¿ [height, width, depth], get_orientation(object_name) -¿ euler angles rotation [rx,ry,rz]. and any other function that you think is necessary to guide the robot (e.g. numpy, scipy, etc).

The guidance function must return a score (float) and a vars_dict (dict). The vars_dict will be used to store the status of conditions relevant to the task com-

pletion. The previous_vars_dict input with contain the vars_dict from the previous iteration. The score must be a continuous value having different values for different states of the robot. States slightly closer to the goal should have slightly higher scores. The next action of the robot will depend on the score returned by the guidance function when queried for many possible future states.

"The state of the robot is a list with 7 elements of the end-effector position, orientation and gripper state [x, y, z, rotation_x, rotation_y, rotation_z, gripper], gripper represents the distance between the two gripper fingers. All distance values are expressed in meters. and the rotation values are expressed in degrees."

start your code with the following import: 'from motor_cortex.common.perception_functions import get_position, get_size, get_orientation'. Do not include any example of the guidance function in the code, only the function itself.

code format example:
'''
from motor_cortex.common.perception_functions import get_position, get_size, get_orientation
# relevant imports
# helper functions
def guidance(state, previous_vars_dict='condition1':False, ...):
# your code here
return score, vars_dict
'''

You are encouraged to break down the task into sub-tasks, and implement helper functions to better organize the code.

You can communicate with a perception_agent and a robotic_agent.
Always indicate who you are talking with by adding 'NEXT: perception_agent' or 'NEXT: robotic_agent' at the end of your message.

Monitor Agent Prompt

You will be given a sequence of frames of a robotic manipulator performing a task, and a guidance code used by the robot to perform the task.
Your job is to describe what the sequence of frames captures, and then list how the robot could better perform the task in a simple and concise way.
Do not provide any code, just describe the task and how it could be improved.

## A.2 GUIDANCE EXAMPLES

The output of the guidance function is a Python script containing high-level abstractions and an understanding of the given tasks. Here we include a guidance code example for an order button press.

Listing 1: Guidance code generated for the button-pushing task in the second iteration. label

```
1 from motor_cortex.common.perception_functions import get_position,
      get_size, get_orientation
2 import numpy as np
3
4 # Helper function to check if the button is pressed
5 def is_button_pressed(button_position, robot_position):
6     distance = np.linalg.norm(np.array(button_position) - np.array(
          robot_position[:3]))
7     return distance < 0.1  # Assuming 0.1m is the threshold for pressing
8
```

```
9   # Main guidance function
10  def guidance(state, previous_vars_dict={'maroon_pressed': False, '
        green_pressed': False, 'azure_pressed': False}):
11      score = 0.0
12      vars_dict = previous_vars_dict.copy()
13
14      # Get positions of the buttons
15      maroon_position = get_position('maroon_button')
16      green_position = get_position('green_button')
17      azure_position = get_position('azure_button')
18
19      # Check the state of the buttons
20      if not vars_dict['maroon_pressed']:
21          if is_button_pressed(maroon_position, state):
22              vars_dict['maroon_pressed'] = True
23              score += 1.0  # Increment score for pressing maroon button
24      elif not vars_dict['green_pressed']:
25          if is_button_pressed(green_position, state):
26              vars_dict['green_pressed'] = True
27              score += 1.0  # Increment score for pressing green button
28      elif not vars_dict['azure_pressed']:
29          if is_button_pressed(azure_position, state):
30              vars_dict['azure_pressed'] = True
31              score += 1.0  # Increment score for pressing azure button
32
33      # Calculate the overall score based on the progress
34      score += (vars_dict['maroon_pressed'] + vars_dict['green_pressed'] +
            vars_dict['azure_pressed']) / 3.0
35
36      return score, vars_dict
```

Listing 2: Guidance code generated for the button-pushing task in the fifth iteration.

```
1   from motor_cortex.common.perception_functions import get_position,
        get_size, get_orientation
2   import numpy as np
3
4   # Helper function to check if the button is pressed
5   def is_button_pressed(button_position, robot_position):
6       distance = np.linalg.norm(np.array(button_position) - np.array(
            robot_position[:3]))
7       return distance < 0.05  # Reduced threshold for pressing
8
9   # Helper function to calculate movement efficiency
10  def calculate_movement_score(current_position, target_position):
11      distance = np.linalg.norm(np.array(target_position) - np.array(
            current_position[:3]))
12      # Penalize for excessive distance
13      if distance > 0.2:  # Arbitrary threshold for excessive distance
14          return -0.5  # Strong penalty for being too far
15      return max(0, 1 - distance)  # Reward for being close
16
17  # Helper function to check orientation using vector mathematics
18  def is_correct_orientation(button_position, robot_orientation):
19      button_vector = np.array(button_position) - np.array([0, 0, 0])  #
            Assuming button position is in world coordinates
20      robot_forward_vector = np.array([np.cos(np.radians(robot_orientation
            [5])),
21                                       np.sin(np.radians(robot_orientation
                                           [5])),
22                                       0])  # Assuming the robot's forward
                                           direction is in the XY plane
23      angle = np.arccos(np.clip(np.dot(button_vector, robot_forward_vector)
            /
```

```
24                                    (np.linalg.norm(button_vector) * np.linalg
                                          .norm(robot_forward_vector)), -1.0,
                                          1.0))
25      return np.degrees(angle) < 15  # Allow 15 degrees of error
26
27 # Main guidance function
28 def guidance(state, previous_vars_dict={'buttons_pressed': {'maroon':
       False, 'green': False, 'azure': False}}):
29      score = 0.0
30      vars_dict = previous_vars_dict.copy()
31
32      # Get positions and orientations of the buttons
33      maroon_position = get_position('maroon_button')
34      green_position = get_position('green_button')
35      azure_position = get_position('azure_button')
36
37      # Get the current robot position and orientation
38      current_position = state
39      current_orientation = get_orientation('robot_end_effector')  #
           Assuming this function exists
40
41      # Button states
42      buttons = {
43          'maroon': maroon_position,
44          'green': green_position,
45          'azure': azure_position
46      }
47
48      # Check the state of the buttons
49      for button, position in buttons.items():
50          if not vars_dict['buttons_pressed'][button]:
51              if is_button_pressed(position, current_position) and
                   is_correct_orientation(position, current_orientation):
52                  # Here, you would implement a feedback mechanism to
                       confirm the button press
53                  # For example: if button_press_successful():
54                  vars_dict['buttons_pressed'][button] = True
55                  score += 1.0  # Increment score for pressing the button
56              else:
57                  score += calculate_movement_score(current_position,
                       position)  # Penalize for distance
58
59      # Check if all buttons are pressed
60      if all(vars_dict['buttons_pressed'].values()):
61          score += 1.0  # Bonus for completing the task
62
63      return score, vars_dict
```

Listing 3: Guidance code generated for the task "push the maroon button, then push the green button, then push the navy button", in iteration 2.

```
1 from motor_cortex.common.perception_functions import get_position,
      get_size, get_orientation
2 import numpy as np
3
4 # Helper functions
5 def distance(point1, point2):
6      return np.linalg.norm(np.array(point1) - np.array(point2))
7
8 def guidance(state, previous_vars_dict={'maroon_pushed': False, '
      green_pushed': False, 'navy_pushed': False}):
9      score = 0.0
10      vars_dict = previous_vars_dict.copy()
11
12      # Get positions of the buttons
```

```python
13    maroon_button_pos = get_position('maroon_button')
14    green_button_pos = get_position('green_button')
15    navy_button_pos = get_position('navy_button')
16
17    # Current end-effector position
18    end_effector_pos = state[:3]
19
20    # Define thresholds
21    push_threshold = 0.05  # 5 cm
22
23    if not vars_dict['maroon_pushed']:
24        # Move towards maroon button
25        dist_to_maroon = distance(end_effector_pos, maroon_button_pos)
26        score = 1.0 - dist_to_maroon  # Closer to the button, higher the
              score
27
28        if dist_to_maroon < push_threshold:
29            vars_dict['maroon_pushed'] = True
30            score += 10  # Bonus for pushing the button
31
32    elif not vars_dict['green_pushed']:
33        # Move towards green button
34        dist_to_green = distance(end_effector_pos, green_button_pos)
35        score = 2.0 - dist_to_green  # Closer to the button, higher the
              score
36
37        if dist_to_green < push_threshold:
38            vars_dict['green_pushed'] = True
39            score += 10  # Bonus for pushing the button
40
41    elif not vars_dict['navy_pushed']:
42        # Move towards navy button
43        dist_to_navy = distance(end_effector_pos, navy_button_pos)
44        score = 3.0 - dist_to_navy  # Closer to the button, higher the
              score
45
46        if dist_to_navy < push_threshold:
47            vars_dict['navy_pushed'] = True
48            score += 10  # Bonus for pushing the button
49
50    return score, vars_dict
```

task: **"press the maroon button, then press the green button, then press the navy button"**

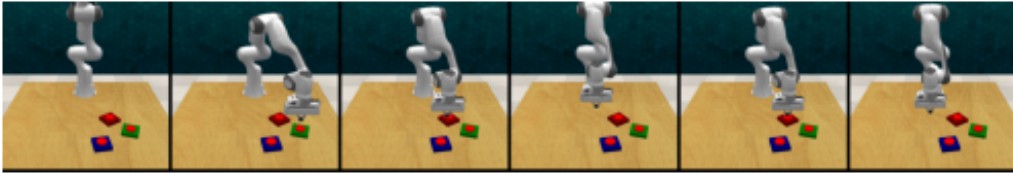

(a) Act3d with no guidance: the policy fails to press the last button (blue), but manages to correctly approach the first 2 buttons reaching them from above with the gripper closed.

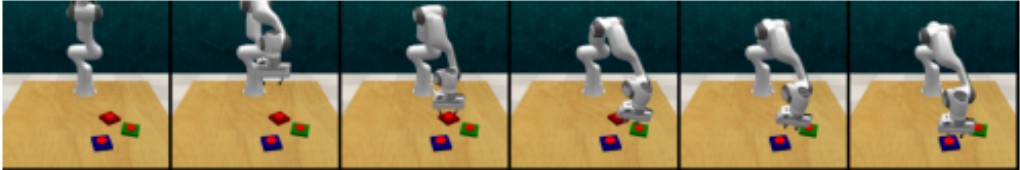

(b) Guidance only (overwriting the base policy): The sequence of movements is correct, but the initial guidance code doesn't account that the buttons should be approached from above.

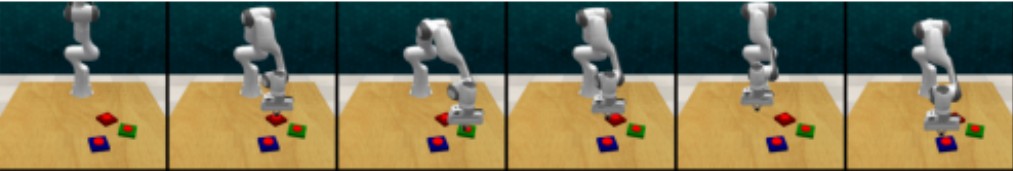

(c) Act3d with 1% guidance: The modified policy captures both the low-level motion of the pre-trained policy and the high-level guidance provided, successfully pressing the sequence of buttons.

Figure 7: Sample rollouts of the guidance correcting a failing task.

