# OpenReview forum: "Grounding Robot Policies with Visuomotor Language Guidance"
_ICLR.cc/2025/Conference — Submitted to ICLR 2025_

### Official Review · Reviewer_EYdK · 2024-10-23

**Soundness:** 2
**Presentation:** 2
**Contribution:** 2
**Rating:** 5
**Confidence:** 4

**Summary:**

This paper proposes an agent-based framework for grounding robotic policies in the current context by taking into account the constraints of both the robot and its environment, using visuomotor-grounded language guidance. The framework consists of a set of VLM-driven agents, each designed for specific roles, with the ultimate goal of improving existing policies.

**Strengths:**

+The paper proposed an agentic robot policy grounding framework that can self improve by generating the action-scoring guidance functions to update the action distribution of a base policy in an online manner.
+Their experiments, conducted in both simulated and real-world robot environments, demonstrate significant performance improvements in adapting existing policies, as well as showcasing zero-shot capabilities.

**Weaknesses:**

- The main contribution of this work appears to be the guidance procedure, which stems from its ability to detect failures. Much of the pipeline focuses on object search using Vision-Language Models (VLM) to guide or refine the action distribution. However, the question arises as to how this approach compares to methods like AHA and RACER, which are instruction-tuned VLMs for failure detection and reasoning. In particular, AHA demonstrates that failure reasoning can improve base policies through a similar guidance mechanism.

- The paper claims that their approach is generalizable across various base policy classes, tasks, and environments. However, in the simulation experiments, evaluations were only conducted on a test set within the same domain distribution as the training data. No evidence was provided to demonstrate generalizability to different tasks or environmental perturbations.

- The authors also suggest that their approach can be used as a zero-shot policy to solve tasks without demonstrations. However, they did not provide any comparisons with other zero-shot methods, such as Manipulate-Anything, VoxPoser, or Code-as-Policies, which would strengthen their claim.

- In Figure 2, the example shows the need for granular object search due to the cluttered environment and large scene. However, this level of detail seems unnecessary for the real-world experiments presented in the paper. It would be beneficial to include more real-world experiments in complex, cluttered environments, where granular object search would be more impactful and better showcase the framework's strengths.

-Duan, Jiafei, Wilbert Pumacay, Nishanth Kumar, Yi Ru Wang, Shulin Tian, Wentao Yuan, Ranjay Krishna, Dieter Fox, Ajay Mandlekar, and Yijie Guo. "AHA: A Vision-Language-Model for Detecting and Reasoning Over Failures in Robotic Manipulation." arXiv preprint arXiv:2410.00371 (2024).

-Dai, Yinpei, Jayjun Lee, Nima Fazeli, and Joyce Chai. "RACER: Rich Language-Guided Failure Recovery Policies for Imitation Learning." arXiv preprint arXiv:2409.14674 (2024).

-Duan, Jiafei, Wentao Yuan, Wilbert Pumacay, Yi Ru Wang, Kiana Ehsani, Dieter Fox, and Ranjay Krishna. "Manipulate-anything: Automating real-world robots using vision-language models." arXiv preprint arXiv:2406.18915 (2024).

-Huang, Wenlong, Chen Wang, Ruohan Zhang, Yunzhu Li, Jiajun Wu, and Li Fei-Fei. "Voxposer: Composable 3d value maps for robotic manipulation with language models." arXiv preprint arXiv:2307.05973 (2023).

-Liang, Jacky, Wenlong Huang, Fei Xia, Peng Xu, Karol Hausman, Brian Ichter, Pete Florence, and Andy Zeng. "Code as policies: Language model programs for embodied control." In 2023 IEEE International Conference on Robotics and Automation (ICRA), pp. 9493-9500. IEEE, 2023.

**Questions:**

- The paper could benefit from a clearer explanation of the system through the figures. Providing more detailed descriptions, especially in figures like Figure 2, would help readers better understand the design principles and flow of the system.

- Additionally, the definitions of "1% and 10% guidance" are unclear and need further clarification. Explaining how these percentages are determined and applied in the system would provide valuable context for understanding their role in guiding the base policy.

-Why is ACT3D with 1% guidance outperforms the 10% guidance variant, it seems counterintuitive.

---

> ### Author Response · Authors · 2024-11-23
>
> * *How does this approach compares to methods like AHA and RACER, which are instruction-tuned VLMs for failure detection and reasoning. In particular, AHA demonstrates that failure reasoning can improve base policies through a similar guidance mechanism.*
>
> **A:** We appreciate the feedback; however, the baselines suggested by the reviewer are still in the pre-print stage and were just recently made available on Arxiv (AHA on 1 Oct 2024, the same day as the ICLR paper deadline) and RACER one week before that). This makes the comparison requests unreasonable for the timeline of this conference. Nevertheless, we will consider these methods in future advances of our work.
>
> * *The paper claims that their approach is generalizable across various base policy classes, tasks, and environments. However, in the simulation experiments, evaluations were only conducted on a test set within the same domain distribution as the training data. No evidence was provided to demonstrate generalizability to different tasks or environmental perturbations*
>
> **A:** By showing that our framework can be used as a zero-shot policy, we demonstrate that it can be used to generate guidance in out-of-distribution cases. Even if the base policies were not exposed to a certain environment or configuration, the system would still be able to provide guidance grounded on trackable elements in the scene.
> It is worth mentioning that the generalizability of our approach is restricted by the capabilities of the VLM used and the precision of the off-the-shelf perception models queried by the agents (GroudingDINO, SAM, AOT). If the environment, objects, or task are out-of-distribution for these models, our framework might struggle to provide meaningful guidance.
>
> * *The authors also suggest that their approach can be used as a zero-shot policy to solve tasks without demonstrations. However, they did not provide any comparisons with other zero-shot methods, such as Manipulate-Anything, VoxPoser, or Code-as-Policies, which would strengthen their claim*
>
> **A:** Even though our method can be applied as a zero-shot policy, our system was designed to also encompass policy grounding and iterative improvement. As such, we adopted design choices that make our framework easily integrate with a diverse set of base policies.  This flexibility makes the direct comparison with dedicated zero-shot methods challenging as they often rely on pre-trained low-level policies, dedicated perception systems, fixed-set actions, or multi-camera setups. Nevertheless, we acknowledge that a deeper comparison with other zero-shot methods could contextualize the zero-shoot capabilities of our framework, and will consider this for future work that explores this facet of our framework.
>
> * *In Figure 2, the example shows the need for granular object search due to the cluttered environment and large scene. However, this level of detail seems unnecessary for the real-world experiments presented in the paper. It would be beneficial to include more real-world experiments in complex, cluttered environments, where granular object search would be more impactful and better showcase the framework's strengths.*
>
> **A:** In Figure 3, we demonstrate how our method benefits from multi-granular detection. In this qualitative example, the target object (white knight) could not be directly detected by the off-the-shelf perception modules. However, by performing the multi-granular search and first locating a “board” and then looking for the “white knight” within the “board” region, our method successfully located the target object and used it to ground the robot's behavior. We updated the paper with a more intuitive and representative figure to demonstrate this use case.
>
> * *The paper could benefit from a clearer explanation of the system through the figures. Providing more detailed descriptions, especially in figures like Figure 2, would help readers better understand the design principles and flow of the system.*
>
> **A:** We appreciate the feedback, we have attached a new version of the paper with updated figures for a clearer presentation of the method.
>
> * *Additionally, the definitions of "1% and 10% guidance" are unclear and need further clarification. Explaining how these percentages are determined and applied in the system would provide valuable context for understanding their role in guiding the base policy.*
>
> **A:** The guidance percentage is defined by the alpha parameter in the equation (3), lines 285 and 286. Its value, between 0 and 1 controls the element-wise weighted average between the base-policy distribution and the distribution generated by our framework. A value of “10% of guidance” translates to an alpha of 0.1, meaning that the logits of each possible action given by the base policy are scaled by 90% and combined 10% of the logits provided by our guidance function.

---

> > ### Author Response · Authors · 2024-11-27
> > **Followup on the rebuttal**
> >
> > With the rebuttal period ending soon, we would like to ask whether our responses have addressed all of your concerns. If so, could you please consider raising your score? Otherwise, we would be happy to respond to any further questions or clarifications.

---

> > > ### Comment · Reviewer_EYdK · 2024-11-27
> > > **Response to author**
> > >
> > > I would like to thank the authors for their response. While some of my concerns have been addressed, there are still a few key points that could further enhance the paper and its claims:
> > >
> > > Formatting Issue: The attached revised PDF has overlapping text, likely due to the use of \vspace in lines 157–159. This should be rectified for clarity.
> > >
> > > Comparison with Prior Work: The authors acknowledge the importance of failure detection and reasoning in guiding policy improvement. However, the REFLECT framework (Liu et al., CoRL 2023), which focuses on failure reasoning, has publicly available code and would be a relevant baseline for comparison.
> > >
> > > Zero-Shot Baseline: The paper claims that the proposed framework enables robots to acquire skills through zero-shot learning or iterative self-improvement (lines 302–303). To strengthen this claim, it would be beneficial to compare the approach with existing zero-shot methods, many of which have RLBench implementations and recovery mechanisms.

---

> > > > ### Author Response · Authors · 2024-12-02
> > > > **2nd Response**
> > > >
> > > > We appreciate the feedback provided by the reviewer. We add further discussion to address these concerns.
> > > >
> > > >
> > > > 1. *Formatting Issue: The attached revised PDF has overlapping text, likely due to the use of \vspace in lines 157–159. This should be rectified for clarity.*
> > > >
> > > > **A**: We appreciate the reviewer highlighting this issue. The latest revision (uploaded on Nov 27) resolves this problem, as well as addressing many of the previously-raised concerns, such as figure captioning and prior work discussion.
> > > >
> > > > 2. *Comparison with Prior Work: The authors acknowledge the importance of failure detection and reasoning in guiding policy improvement. However, the REFLECT framework (Liu et al., CoRL 2023), which focuses on failure reasoning, has publicly available code and would be a relevant baseline for comparison.*
> > > >
> > > > **A**: While REFLECT appears to pursue similar goals, its focus diverges significantly from our framework in several key aspects, preventing it from being a relevant baseline comparison:
> > > >
> > > > * REFLECT doesn't propose its framework for generalized failure correction (see Section 3.3 in their paper). Their approach is exclusively focused on generating captions/explanations and producing a dataset of explanations. Meanwhile, our overall framework is geared towards modifying policy behavior, where failure-detection comprises only one of the modules within our policy-correction system. Work such as REFLECT could be used to further optimize the Monitor Agent component of the g-MotorCortex framework, which we reserve for future work.
> > > >
> > > > * REFLECT reuses the approach from "Language models as zero-shot planners" by taking a sentence embedding to map the captions to pre-existing task primitives from the AI2THOR environment. Our guidance framework, on the other hand, is not conditioned on any single environment, specific policy, or action space—thereby achieving a broader scope in a more flexible and adaptable way. We support this claim by running our framework both in simulation as well as real-robot experiments, as well as with regression and classification policy types.
> > > >
> > > > 3. *Zero-Shot Baseline: The paper claims that the proposed framework enables robots to acquire skills through zero-shot learning or iterative self-improvement (lines 302–303). To strengthen this claim, it would be beneficial to compare the approach with existing zero-shot methods, many of which have RLBench implementations and recovery mechanisms.*
> > > >
> > > > **A**: As a reminder, our framework is designed to work as guidance on top of base policies, by iteratively shifting their action distributions. Our experiments showcase that, even in extreme cases where the base policies are not reliable (e.g., a completely untrained or randomly-initialized policy), we are able to learn and inject important concepts for completing the tasks, in a flexible manner, accommodating policies with different action spaces and output formats.
> > > >
> > > > Our framework is not designed to function as a stand-alone zero-shot approach, nor does it aim to surpass state-of-the-art zero-shot methods in manipulation, but rather complement existing zero-shot policies with grounded corrective guidance; the strength of our framework lies in iteratively enhancing behavior (and, by extension, success rates) over time. We refer the reviewer to Figure 4, which demonstrates the progressive improvement achieved through successive iterations of our framework.
> > > >
> > > > ---
> > > >
> > > > Overall, we thank the reviewer for the feedback. We address distinctions of our work with prior approaches in detail in the Related Work section (lines 138-140); we will further augment this section in the final version to make the comparisons more clear.
> > > >
> > > > Please let us know if this now satisfies all of the reviewer’s concerns and, if so, we would appreciate it if the reviewer would consider increasing their score.

---

### Official Review · Reviewer_itkp · 2024-10-30

**Soundness:** 2
**Presentation:** 2
**Contribution:** 3
**Rating:** 5
**Confidence:** 4

**Summary:**

The authors propose a method to guide robot policy with the guidance of several Vision Language Models (VLM). Specifically, the authors use several VLMs to compute scores of actions during execution and use the score to improve/correct the policy. The authors evaluate their approach on the RLBench and real world tasks and show improved performance.

**Strengths:**

1. The high-level idea of this paper is interesting. Extracting VLM's knowledge to guide low level behavior is an interesting direction.
2. In the experiments, the authors evaluate their method on several realistic tasks and demonstrate the effectiveness of applying VLM guidance.

**Weaknesses:**

Major Weaknesses: Clarity of Presentation

The main weakness of this paper is the lack of clarity in presenting its concepts and methods. Many key ideas are not explained in sufficient detail, which makes it difficult to follow the proposed approach.

1. In Figure 2, numerous arrows connect concepts like Robotic Agent, Advisory Agent, and Grounding Agent, yet these elements are absent from Algorithm 1. Without these agents and connections represented in the algorithm, understanding the overall inference procedure becomes challenging. A clearer, more detailed explanation of the full inference process would be beneficial.

2. In Algorithm 1, it is unclear how to implement the "Generate_Guidance" and "Infer future state" steps in Line 9.

3. It appears that the authors sample action $a$ from a product distribution $\pi (a|o) * G(a|o, ...)$. This approach is challenging, particularly with complex probabilistic distributions. What are the parameterizations of $\pi$ and $G$? In experiments, it looks like $\pi(a|o)$ is primarily a diffusion-based policy. How is the computed score then used to guide diffusion sampling? Additionally, is $G_{\pi_t}$ or $G(a|o, ...)$ represented as a vector?

4. How is "iterative improvement" achieved, as referenced in Line 300 and Figure 4?

5. Figures 3 and 5 are difficult to interpret. The lack of clarity around score representation (as mentioned in point 3) may contribute to the difficulty in understanding these figures.

**Questions:**

See the weaknesses part. I would like to raise my score if the authors can resolve the clarity issues by more details and convince me the effectiveness of their approach.

---

> ### Author Response · Authors · 2024-11-23
> **Response [1/2]**
>
> * *A detailed explanation of the full inference process*
>
> **A:** We appreciate the feedback. We included an updated version of the paper with a clearer representation of the process of guidance generation.  We represent graphically how each agent interacts with each other to generate the guidance function, a process referred to as Generate_Guidance$(o_t, s_{t+1})$  in the algorithm I. We provide a summary of this process, already described in Section 3.3 and further clarified in the revision of the paper: Given an image of the environment and a robot state, the group of self-orchestrated agents aims to generate a guidance function (Python code) to modify the action distribution of a base policy. Each agent is a conversational Visual Language Model (VLM) prompted to exercise a role. These agents use tools to enhance their responses through function calling. The agents interact in a “group chat” fashion, sequentially via natural language. The order of speakers is defined by the agents. Figure 2 and lines 198-225 depict the flow of information exchanged between these agents during the guidance generation procedure. The specific content of the messages exchanged and the order of speakers is flexible, however, the agent's prompt outlines an overall procedure to be followed. The diagram in a new figure added in the Appendix depicts this procedure. The “Advisor agent” coordinates the process and is responsible for the code generation; the “grounding agent” can interact with perception models and is responsible for detecting, segmenting, and tracking elements in the scene to ground the code on existing elements in the scene; The “robotic agent” ensures that the code satisfies the platform restrictions and the expected input/output formats; The “Monitoring agent” is used to analyze the execution of the robot in the environment and raise possible reasons of failure.
>
> * *How are Generate_Guidance and Infer future state implemented?*
>
> **A:** The algorithm proposed leverages the Agentic Framework to generate the “guidance code”, which then alters the action probabilities of the base policy. The Agents are instantiations of VLMs empowered with tool-using capabilities. During the guidance code generation, these agents exchange several messages with each other (in natural language) in a self-orchestrated manner. With that, defining on their own the content and recipient agent of every message exchanged until the guidance code is considered to be adequate for the robot, task, and current state of the environment. This process is explained and exemplified in sections 3.2 and 3.3. The process of inferring future states is given by the Dynamics model of the robot, inferring future robot states from current state and action. This process is highly dependent on the robotic embodiment used and the action space of the base policy. For our experiments, the action and state space of the policy match, meaning that the policy outputs the poses that the robot should be moved to, and its gripper state. This simplifies our formulation to a case where the possible future states of the robot are directly obtained by the possible actions that the policy can output.
>
> ... (continue in response 2/2 )

---

> > ### Author Response · Authors · 2024-11-23
> > **Response [2/2]**
> >
> > * *Parameterizations of $\pi$ and $G$*
> >
> > **A:** The guidance is parametrized in the same space as the output of the policy. In our experiments, this action space is SE(3) x gripper $\[0,1\]$, which is represented as a continuous Cartesian coordinate for the poses of the end-effector and a value $\in$ [0,1] for the gripper state. Section “Action-space Adaptation” (line 273) explains the process required to match the parameterization of the guidance and base policy.
> >
> > * *How is the score used to guide diffusion sampling?*
> >
> > **A:** Our system was evaluated on classification and regression policies. The diffusion-based policy (3D Chained Diffuser) outputs a single value for each component of the action space, configuring it as a regression policy. As described in Section 4.1 (line 354), we assume a distribution around the single predictions of the regression policies. For instance, the policy’s single output, expressed as vector ($o \in R^7$), becomes a Gaussian distribution centered in the output ( $N_7(o, \sigma)$ ). This inferred distribution is sampled $n$ times, where each sample is associated with its respective logit value. For each possible action sampled, a guidance score $G_{\pi}$ is computed using the guidance function followed by a normalization. These samples of guidance distribution are then combined with the logits of the original $n$ sampled actions using an element-wise weighted average. This process is described in section 3.4 and Algorithm 1.
> >
> > * *How is iterative improvement achieved?*
> >
> > **A**: The Iterative improvement is achieved by updating the guidance function for the failure cases. In detail, for a given task and environment setup, our group of self-orchestrated agents (VLMs + tools) generates a guidance function. The function is used to modify the action distribution of a base-policy and a sampled action is used to act in the environment (Algorithm I). If the robot fails to perform the task, the monitoring agent tries to identify the causes of the failure by analyzing key frames of the rollout and previous guidance code (line 220). The possible causes of the failure are used as feedback to the other agents, which will then iterate over the guidance code to improve it until the robot succeeds in the task.
> >
> > ---
> >
> > Overall, we thank the reviewer for suggestions that improve the clarity and presentation of our paper. We made some changes to facilitate comprehension. We hope that this response and paper updates address all your concerns about our work. If that is the case, we look forward to an update on your rating score.
> >
> > Best regards.

---

> ### Comment · Reviewer_itkp · 2024-11-25
> **Thank you for your response.**
>
> I would like to thank the authors for their clarifications. After reviewing the response, I still have a few concerns as outlined below:
>
> 1. Coupling with a specific robot policy: The proposed approach appears to be tightly coupled with a specific robot policy (3D Chained Diffuser), which utilizes a high-level macro action to guide a low-level trajectory generation policy. The VLM guidance is implemented by modifying the distribution of intermediate macro actions based on VLM outputs (the proposed method itself is also computationally expensive). However, it remains unclear how this approach could be adapted to more general robot policies that use standard action spaces and do not have a hierarchical structure (e.g., diffusion policy, ACT, etc.). There is limited insight into the algorithmic aspects of integrating VLM guidance into such more general settings.
>
> 2. Comparison to related work: On the system side, the improvement seems primarily driven by VLM-based object understanding (e.g., detection). Related works, such as OK-Robot [1] and VoxPoser [2], also employ VLMs to ground language to objects and use specific robot policies (e.g., AnyGrasp) to manipulate those objects. It is unclear how the proposed method addresses the limitations of these existing approaches or advances beyond them in a meaningful way (e.g. the proposed method can guide a specialized robot manipulation policy (with few assumptions on its architecture) to perform better in challenging dexterous/precise manipulation tasks.).
>
> Given the authors' efforts to clarify certain aspects, I am increasing my score to 5. However, I believe the concerns outlined above merit further discussion and exploration.
>
> [1] Liu et al. OK-Robot: What Really Matters in Integrating Open-Knowledge Models for Robotics. In RSS 2024.
> [2] Huang et al. VoxPoser: Composable 3D Value Maps for Robotic Manipulation with Language Models. In CoRL 2023.

---

> > ### Author Response · Authors · 2024-11-27
> > **2nd Response [1/2]**
> >
> > We appreciate your feedback and score increase. Indeed, these discussion points are of great importance for the understanding of our contribution, we address them deeper below and emphasize them in an updated version of the paper. We hope these further clarifications and discussions will ease your remaining concerns.
> >
> > -  _1. Coupling with a specific robot policy: The proposed approach appears to be tightly coupled with a specific robot policy (3D Chained Diffuser), which utilizes a high-level macro action to guide a low-level trajectory generation policy. The VLM guidance is implemented by modifying the distribution of intermediate macro actions based on VLM outputs (the proposed method itself is also computationally expensive). However, it remains unclear how this approach could be adapted to more general robot policies that use standard action spaces and do not have a hierarchical structure (e.g., diffusion policy, ACT, etc.). There is limited insight into the algorithmic aspects of integrating VLM guidance into such more general settings._
> >
> > **A:**
> >
> > Oh, there must be some misunderstanding! As a reminder, our framework is designed to be agnostic of the specific policy architecture, in order to accommodate diverse policy classes; it modifies only the posterior distribution of the base policies. As mentioned in line 156 (170 in the previous version) and described in section 3.4, we take as an assumption that a nominal dynamics model (D) of the robot is available; such a model, commonly available in many robotic systems, allows the inference of future possible states of the robot given its current state and possible actions (algorithm 1, line 9). These design choices make the framework independent of the action space representation and the specifics of the robot form-factor.
> >
> > Nevertheless, it is true that it is important to consider the type of output that the policy provides. As described in section 3.4, we do this as follows: **[Classification policies]** if the policy outputs a distribution of probabilities (or logits) across a set of possible actions, the guidance scores (computed on the inferred possible future states) can be directly used to change the action distribution and therefore the final action executed by the robot. **[Regression policies]** On the other hand, if the policy outputs a single value for each dimension of the action space, the distribution of the output needs to be inferred and sampled—thereby allowing inference of possible future state, followed by the computation of guidance scores and the modification of the policy output.
> >
> > Considering the expensive computational costs of VLMs, we also designed our system to use these powerful models detached from the action loop, while still leveraging their knowledge and perception capabilities. A few design choices allow us to guide the policies during inference time without introducing considerable latency:
> >
> > 1. Relevant objects are tracked in real-time using AOT. The initial detection and segmentation are made by the Grounding Agent + perception models (GroundingDINO and SAM) before the robot starts executing the task, and are only recomputed if the tracking is temporarily lost.
> > 2. The guidance is expressed as quickly executable code functions, grounded on the trackable elements in the scene, which allows the guidance scores to be computed for thousands of states in a fraction of a second.
> > 3. Our system was implemented in a threaded manner, allowing the base policies to run concurrently with the tracking, detection, and code generation.
> > 4. The use of “redis” to interface the codebase of existing policies and the guidance procedures allows easy and low-latency integration with our guidance system.

---

> > > ### Author Response · Authors · 2024-11-27
> > > **2nd Response [2/2]**
> > >
> > > - *Comparison to related work: On the system side, the improvement seems primarily driven by VLM-based object understanding (e.g., detection). Related works, such as OK-Robot [1] and VoxPoser [2], also employ VLMs to ground language to objects and use specific robot policies (e.g., AnyGrasp) to manipulate those objects. It is unclear how the proposed method addresses the limitations of these existing approaches or advances beyond them in a meaningful way (e.g. the proposed method can guide a specialized robot manipulation policy (with few assumptions on its architecture) to perform better in challenging dexterous/precise manipulation tasks.).*
> > >
> > > **A:**
> > >
> > > Leveraging open-vocabulary VLM for object and scene understanding is an approach already explored by a few works. However, our framework combines this concept with key design choices to tackle a broader scope of problems. For instance, our work presents a few fundamental differences, compared with VoxPoser and OK-robot. Beyond the zero-shot capabilities of our approach (learning new skills from scratch), our approach proposes modifying the action distribution of an arbitrary base policy, enabling the correction of errors and the improvement of existing models without the need of fine-tuning or collecting extra data. This process of policy guidance surpasses the scope of VoxPoser and OK-robot. To our knowledge, we are the first to propose such a formulation.
> > >
> > > Furthermore, our method addresses two main limitations that are acknowledged by the VoxPoser and OK-robot papers:
> > >
> > > 1. Granularity problem on off-the-shelf perception models:
> > >
> > > As mentioned in VoxPoser, detecting fine-grained object geometries can be challenging when using off-the-shelf perception models in a single-step zero-shot manner; tuning the model’s hyperparameter and queries to accommodate multiple levels of details becomes a problem for such methods. We mitigate this problem by proposing a multi-granular object search (grounding agent), where we allow a conversational VLM to search for objects in a flexible, iterative, and recursive manner. For instance, our agent is able to query open-vocabulary models with semantically similar object names, look for possible parent objects present in the scene, and repeat the searches on the crops of the image. As we demonstrate in Section 4.2 and Figure 3, these improvements reduce the need for manually tuning the grounding models and enable better grounding with the already-available models. As a reminder, further details can be found in the updated Appendix in the revision.
> > >
> > > 2. Failure correction:
> > >
> > > Our system addresses the key limitation of failure-correction by introducing an iterative improvement loop. The Monitor Agent leverages visual feedback from previous rollouts, allowing the update of guidance codes to address failure cases. This process is key to employing the solution for a diverse range of environments and tasks and is crucial for correcting faulty policies which, themselves, present distinct sorts of biases and need different types of guidance. In OK-robot, this correction process is not addressed and is raised as a limitation. In Voxposer, the corrections rely on new language instructions to be provided, in the form of costly direct human input. Our experiment on learning robot skills from scratch (Figure 4) demonstrates how such iterative and automatic improvement can modify guidance codes to achieve better task completion, without the need of any pre-trained low-level policies or additional human feedback.

---

> > ### Author Response · Authors · 2024-12-02
> > **Followup on rebuttal**
> >
> > With the rebuttal period ending soon, we would like to ask whether our responses have addressed all of your concerns. If so, could you please consider raising your score? Otherwise, we would be happy to respond to any further questions or clarifications.

---

### Official Review · Reviewer_KDxu · 2024-11-04

**Soundness:** 3
**Presentation:** 3
**Contribution:** 3
**Rating:** 6
**Confidence:** 2

**Summary:**

The paper proposes g-MotorCortex, an agentic robot policy grounding framework that can self improve by generating the action-scoring guidance functions to update the action distribution of a base policy in an online manner. The paper presents experiments in both simulated and real-world robot settings and shows significant performance improvements on both adapting existing policies and learning new skills from scratch.
The proposed method is shown to be robust against cluttered and unseen environments, thanks to multi-granular object search, which enables flexible visuomotor grounding.
The authors also open source the code, models, and system prompts.

**Strengths:**

The paper is clearly written and well structured. It is easy to read and to follow through the contributions and components of the proposed method.
The experimental results are well presented as well and span across different axes of evaluations, which make it easier to appreciate the proposed method and contributions.
The results presented are interesting and significant to the field, and well demonstrate a nice integration of VLMs with robot control.

**Weaknesses:**

The experimental results show performance of three algorithms: Act3D, 3D Diffuser Actor, and a RandomPolicy. The reasons behind this choice is not clear and it would be helpful to understand what's behind this, and whether other types of algorithms would also be suitable and how their performance is expected to be.
The real robot evaluations is interesting, although the task defined is not interactive at a physical level, being only a reaching task. The results are interesting and valuable, however real-robot challenges arise often when interaction with environments exist.

**Questions:**

- Can you clarify the choice of algorithms chosen? Are results available from other algorithms too? Are there specific reasons behind this choice?
- What are the limitations of the learning from scratch? Can the system effectively incorporate preferences/demonstrations?
- Can you elaborate on why the self-improvement experiments plateau?

---

> ### Author Response · Authors · 2024-11-23
>
> We sincerely thank you for the positive rating and feedback. In response to your comments, we provided explanations to address them as well as revisions to our paper to aid in clarity. We hope to highlight the strengths of our framework, increasing your confidence in its contribution to the field. Thank you once again for your constructive review and valuable insights.
>
> 1. *The experimental results show performance of three algorithms: Act3D, 3D Diffuser Actor, and a RandomPolicy. The reasons behind this choice is not clear and it would be helpful to understand what's behind this, and whether other types of algorithms would also be suitable, and how their performance is expected to be*.
>
> **A**: We choose baseline policies that showcase diversity in representation and achieve strong performance in the language-conditioned manipulation field. The distinct output formats, and policy architectures, showcase that our framework could be adaptable to a wider range of policies that follow similar specifications. Section 3.4 discusses how our framework could be suitable for a wider range of policies. The random policy showcases that our framework can guide even untrained policies in diverse environments.
>
> 2. *The real robot evaluations is interesting, although the task defined is not interactive at a physical level, being only a reaching task. The results are interesting and valuable, however, real-robot challenges arise often when interaction with environments exists.*
>
> **A**: We showcase tasks that require a complex understanding of the scene, e.g., pressing buttons in a specific order and reaching for a white knight in a cluttered environment. Both of these tasks, although relatively simple in their low-level motion, are challenging due to the fine-grained perception and long-term reasoning required. We also clarify that the button-pressing task includes physical interaction with the button.
>
> 3. *Can you clarify the choice of algorithms chosen? Are results available from other algorithms too? Are there specific reasons behind this choice?*
>
> **A**: The chosen base policies (Act3D and 3D diffuser actor)  are strong baselines in the field of language-conditioned manipulation. These approaches are fundamentally different in their output formats (classification vs. regression) and architecture; they represent informative model classes that act as great use cases for illustrating the versatility of our policy guidance framework. By improving the performance of both algorithms, we demonstrated that our framework can address and improve a diverse set of policies. The performance of these methods is reported in Table 1.
>
> 4. *What are the limitations of the learning from scratch? Can the system effectively incorporate preferences/demonstrations?*
>
> **A:** The main limitations of the learning-from-scratch process are based on the need to improve upon motions that require precise low-level control of the robot or reasoning about fine-grained interactions with the scene (e.g., areas of contact). Crucially, these limitations are mainly explained by limits on the current capabilities of the grounding models (GroundingDino, SAM, and AOT) and challenges in the ability for Visual Language Models (VLMs) to perform time-dependent scene understanding. Still, even with the current capabilities of these grounding models and VLMs, we are able to show increments of performance improvements on different tasks.
>
> 5. *Can you elaborate on why the self-improvement experiments plateau?*
>
> **A:** Due to the intrinsic limitations of the perception models used and the scene understanding of the monitoring agent, our ability to detect and correct failures becomes restricted. Because of that, our self-improvement experiments reach a plateau in performance. A future direction we are willing to explore is how to combine the self-improvement approach proposed with other exploration methods such as Curiosity-driven RL. Combining both could lead to improvements in the low-level control of the robot while allowing sample efficiency by targeting the exploration with our system’s guidance.

---

> > ### Comment · Reviewer_KDxu · 2024-11-25
> >
> > Thank you for addressing my comments and providing the clarifications. I appreciate how the other reviewers' concerns have been addressed, though there may be still a few points that may need further attention.

---

### Author Response · Authors · 2024-12-03
**General Response and Rebuttal Summary**

We thank the reviewers for the valuable feedback and discussion. For clarity, we include a summary of the changes made to the revision.

- **Clarity and Presentation:** We revised the paper to improve clarity, including updated figures (e.g., **Figure 2**) to better illustrate the guidance procedure and agent interactions. **Algorithm 1** now aligns more clearly with the described processes, and additional implementation details are provided in **Appendix A.1**. These changes better elucidate how our policy grounding framework can improve existing policies (see **Table 1**) and guide them to success even in out-of-distribution cases (**Figure 4**). Furthermore, it clarifies how the group of self-orchestrated agents leverages the multi-granular object search proposed to provide meaningful guidance even in cluttered environments (**Figure 3** and **Section 3.3**).

- **Baseline Comparisons:** We addressed concerns regarding the baselines and provided justifications for the chosen algorithms (Act3D, 3D Diffuser Actor, and RandomPolicy) as diverse and representative use cases for our framework. While other methods like REFLECT and dedicated zero-shot approaches were acknowledged, we clarified their limited relevance to the overall scope of our approach. Our response helps better situate our framework among prior works, showcasing two major key differences: flexible and actionable policy correction, as well as zero-shot capabilities tailored to the base policies’ action space (**Figure 4**).

-  **Iterative Improvement Clarification:** We highlight that the framework incrementally enhances task success rates, even in challenging scenarios with unreliable (untrained) base policies, by leveraging feedback from the perceived robot’s behavior. In comments to reviewers, we further clarified our system's limitations originating in its off-the-shelf components and elucidated the reason behind learning plateaus in our iterative improvement experiments.

We once again thank the reviewers for their insightful comments. We believe that our responses addressed in a meaningful way the raised concerns and clarified the strengths of our work.

---

### Meta-Review · Area_Chair_Lx8S · 2024-12-20

**Metareview:**

The paper introduces a framework leveraging Visuomotor Language Models (VLMs) to guide robot manipulation policies. The proposed system demonstrates promising performance in adapting policies and enabling zero-shot task execution. While reviewers recognized the novelty and potential impact of this work, significant concerns were raised regarding clarity, presentation, and experimental evaluation.

Strengths:

- The paper addresses a compelling challenge at the intersection of robotics and foundation models, presenting a novel framework for dynamic policy guidance.

- Experiments demonstrate promising results in both simulation and real-world environments.

- The modular, agent-based design enhances system flexibility and usability across diverse tasks and policies.

Weaknesses:

- Reviewers consistently pointed to unclear explanations of key concepts (e.g., guidance generation, inference procedures). Updated figures and algorithmic clarifications partially addressed these concerns.

- The citation and references to prior work on was seen as insufficient, particularly given the claims of generalizability and zero-shot learning. I would like to encourage the authors to have a thorough review of recent works on robotic control and planning methods using VLMs, including but not limited to the papers mentioned by the reviewers.

- Real-world experiments are limited to simpler tasks, lacking tests in more cluttered and challenging settings where the method's strengths would be better highlighted.

Although I agree that the paper could be significantly improved with a more comprehensive discussion of prior work on VLM-based robotic control, I would like to point out that Reviewer EYdK's request for comparison to concurrent work such as AHA and RACER is unnecessary and unfair. Notably, AHA is a concurrent submission to ICLR 2025 ([Submission 506](https://openreview.net/forum?id=JVkdSi7Ekg)) and was first arXived on October 1st, 2024, making such a request impractical and against conference guidelines. Therefore, this request was not considered in the final decision on the paper.

While the paper's novelty and potential are apparent, the concerns regarding clarity and insufficient discussion on prior work weaken its overall contribution in its current form. Given these considerations, I recommend rejection for ICLR 2025, with encouragement to improve clarity, expand experimental validation, and engage more comprehensively with related work in future submissions.

**Additional Comments On Reviewer Discussion:**

During the rebuttal, reviewers highlighted concerns about clarity, insufficient comparisons with prior works, and limited experimental scope. The authors improved the paper’s clarity and figures and addressed some limitations, but concerns about lack of discussion about prior work and generalizability remained unresolved. While the authors’ efforts were appreciated, the remaining gaps in validation and prior work discussion led to a recommendation for rejection, with encouragement for further refinement.

---

### Decision · Program_Chairs · 2025-01-22

Reject